# Linearly Decomposing and Recomposing Vision Transformers for Diverse-Scale Models

**Shuxia Lin**[1,2][*], **Miaosen Zhang**[1,2][*], **Ruiming Chen**[1,2][*], **Qiufeng Wang**[1,2], **Xu Yang**[1,2][†], and **Xin Geng**[1,2][†]

[1]School of Computer Science and Engineering, Southeast University, Nanjing 210096, China
[2]Key Laboratory of New Generation Artificial Intelligence Technology and Its Interdisciplinary Applications, Southeast University, Ministry of Education, China
`{shuxialin, 230228501, 220232251, qfwang, xuyang_palm, xgeng}@seu.edu.cn`

## Abstract

Vision Transformers (ViTs) are widely used in a variety of applications, while they usually have a fixed architecture that may not match the varying computational resources of different deployment environments. Thus, it is necessary to adapt ViT architectures to devices with diverse computational overheads to achieve an accuracy-efficient trade-off. This concept is consistent with the motivation behind **Learngene**. To achieve this, inspired by polynomial decomposition in calculus, where a function can be approximated by linearly combining several basic components, we propose to linearly decompose the ViT model into a set of components called learngenes during element-wise training. These learngenes can then be recomposed into differently scaled, pre-initialized models to satisfy different computational resource constraints. Such a decomposition-recomposition strategy provides an economical and flexible approach to generating different scales of ViT models for different deployment scenarios. Compared to model compression or training from scratch, which require to repeatedly train on large datasets for diverse-scale models, such strategy reduces computational costs since it only requires to train on large datasets once. Extensive experiments are used to validate the effectiveness of our method: ViTs can be decomposed and the decomposed learngenes can be recomposed into diverse-scale ViTs, which can achieve comparable or better performance compared to traditional model compression and pre-training methods. The code for our experiments is available in the supplemental material.

## 1 Introduction

The pre-training Vision Transformers (ViTs) [13] have become fundamental to various applications, including image classification [59, 7], object detection [29, 52], semantic segmentation [46, 67], and multimodal tasks [63, 58]. However, these ViTs typically have a standard and relatively fixed architecture, which poses challenges for deployment in diverse real-world settings, i.e., devices in different application scenarios have different computational capabilities, making standard, fixed-size ViTs unsuitable for direct deployment. Furthermore, the storage demands of a traditional ViT may exceed the capabilities of certain devices. For instance, the memory size for the ViT-B/16 model [13] with 86M parameters is approximately 320 MB, while some devices may find this prohibitive due to inherent physical constraints, e.g., the limited RAM capacity of the Raspberry Pi 1 with 256MB-512MB of RAM.

---

[*]These authors contributed equally.
[†]Corresponding authors.

38th Conference on Neural Information Processing Systems (NeurIPS 2024).

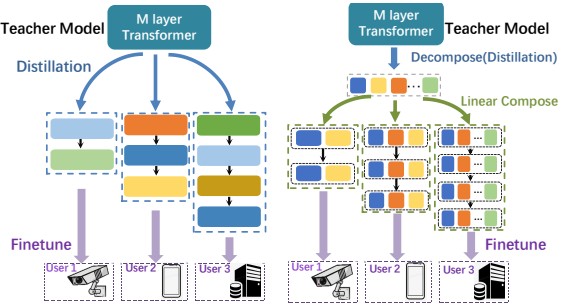
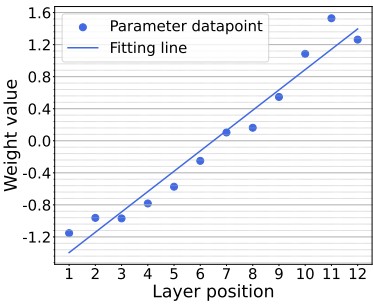

Figure 1: Comparison between Knowledge Distillation (KD) (left) and our method (right). KD requires $N$ times of training with the teacher model on data to produce $N$ different scale models for $N$ clients. In contrast, our method requires only a single training to decompose the teacher model into different learngenes that can be economically and flexibly recomposed into models with diverse layers to meet different client needs.

Figure 2: After using PCA to reduce the dimension of the parameters of a well-trained ViT, we find that the parameters of the most layers have an approximately linear correlation with their layer position. More details are given in the supplementary material.

To address this limitation, researchers have developed various compression techniques [6, 62, 55, 28] to reduce the model size to make them suitable for the application scenarios while meantime maximum preserving the model ability. Typical approaches including micro-architecture design[23], model pruning[14, 68], quantization[43], and knowledge distillation[62]. However, in these approaches, the architecture of the smaller model, such as the number of layers, is predetermined and thus lacks the flexibility to meet the diverse needs of real-world deployment scenarios. Consider the scenario depicted in Figure 1: to obtain $N$ models of different sizes for different devices, the knowledge distillation methods need to go through $N$ separate trainings. Striking a balance between flexibility and accuracy is challenging and often requires complex training strategies to ensure satisfactory model performance.

Then we may ask, is there a way to more flexibly and economically generate $N$ smaller models from a large one, while maintaining the performance of the smaller ones? In calculus, we know that a function can be approximated by linearly combining several basic components with corresponding weighted coefficients at various levels of precision, e.g., Taylor series [45], Chebyshev Polynomial of the First Kind [31], Fourier series [47] and so on. Such a mathematical theorem induces two engineering insights: (1) a function can be decomposed into a series of basic components, i.e., the polynomial terms, and (2) we can combine an appropriate number of the basic components to approximate a function with a certain degree of precision. Motivated by these insights, we explore whether a large ViT can be incrementally decomposed into basic components, which we term "learngenes", mimicking the behavior of organismal genes as proposed by [49, 51]. In their innovative learning framework, critical knowledge is continuously condensed as learngenes during the evolution of the ancestry model, which are then inherited to initialize descendant models of varying sizes. Similarly, in our approach, we propose to decompose a large ViT into a set of learngenes, encapsulating critical knowledge from the original model. These learngenes can then be flexibly recomposed to initialize ViT models of different scales, adapting to diverse deployment scenarios.

Specifically, to achieve this, we propose that each layer $\boldsymbol{W}$ of the ViT can be decomposed into some basic learngenes: $\boldsymbol{A}_1, ..., \boldsymbol{A}_N$, where each $\boldsymbol{A}_n$ includes all submodules in $\boldsymbol{W}$, such as Multiheaded Self-Attention (MSA), Multi-Layer Perceptron (MLP), Feed Forward Networks (FFN), and so on. However, a ViT contains many layers and if we specify a learngene space for each layer, then an $L$-layer ViT requires a total of $N \times L$ basic learngenes, which still requires a large amount of computation to decompose. Interestingly, as shown in Figure 2 of [54], the parameters of most layers in the well-trained ViT of [13] have an approximately non-decreasing trend after PCA dimension reduction. Among several possible fitting functions, the linear function is used here for its simplicity and effectiveness in approximating this trend. Thus we further assume that different layers of a ViT can also be linearly decomposed into the same learngene space, which means that we can also share the decomposed learngene across different layers in recomposition.

To summarize, we assume that the parameters of the $l$-th layer can be got by

$$\boldsymbol{W}_l = \sum_{n=1}^{N} a(l, n) \times \boldsymbol{A}_n, \tag{1}$$

where $a(l, n)$ are the linear combination coefficients, which are pre-defined and depend on the layer and learngene. Here, "pre-defined" denotes that $a(l, n)$ satisfies a pre-defined role, e.g., we find that the first-kind Chebyshev polynomial formula is a suitable one, and does not require adjustment during training. Also, when recomposing, given the layer rank $l$ and which learngenes should be used, we can first compute the values of $a(l, n)$ and then directly use the Equation 1 to construct the parameters of a ViT with any layer.

Such decomposition-recomposition strategy has the following two distinctive characteristics. First, such a decomposition strategy provides us with a novel training mechanism that allows us to iteratively train each learngene $\boldsymbol{A}_n$ in an incremental way. Specifically, to train the first learngene $\boldsymbol{A}_1$, we train a ViT where the parameter of the $l$-th layer is $\boldsymbol{W}_l = a(l, 1) \times \boldsymbol{A}_1$. Then we begin to train the second learngene $\boldsymbol{A}_2$ by constructing a ViT where $\boldsymbol{W}_l = a(l, 1) \times \boldsymbol{A}_1 + a(l, 2) \times \boldsymbol{A}_2$. During training, $\boldsymbol{A}_1$ is fixed and only the parameters of $\boldsymbol{A}_2$ are updated. Then such a process is iterated a few times until all learngenes are trained. Compared to traditional pre-training methods, the decomposition approach not only generates a series of learngenes, but also yields a variety of pre-trained models with different scales during training. Importantly, the decomposed model achieves comparable performance to the pre-training method under the same parameters and training epochs.

Second, after decomposition, the decomposed learngenes can be flexibly recomposed into diverse scale ViTs with different depths without training. Meanwhile, we can use only an appropriate number of learngenes instead of all of them for recomposition to achieve dynamic accuracy-efficiency trade-offs. For example, we can use the previous 6 learngenes $\{\boldsymbol{A}_1, ..., \boldsymbol{A}_6\}$ to recompose a 4-layer ViT where the $k$-th layer is initialized as $\boldsymbol{W}_k = \sum_{n=1}^{6} a(k, n) \times \boldsymbol{A}_n$, where $a(k, n)$ is still pre-defined by satisfying a specific role, e.g., the first-kind Chebyshev polynomial coefficients. In contrast to model compression methods where each training only targets a specific model size, which is not flexible enough to meet the needs of real-world deployment scenarios.

Finally, our experiments demonstrate the viability of linear decomposition and subsequent linear recomposition for ViT models. During decomposition, we observe that incrementally increasing the number of learngenes allows the performance to match that of a classically trained, integral ViT model. In recomposition, we can flexibly use an appropriate number of decomposed learngenes to construct ViTs of different layers. This method achieves performance comparable to other model compression methods, while facilitating the economical generation of models for diverse client requirements, e.g., reducing the training cost by $80\times$ when generating the same number of models with diverse scales and initializations.

## 2 Related work

In this paper, we present a method that decomposes the parameters of the ViT model into a series of learngenes combined by polynomial coefficients. Then, the decomposed learngenes are used to flexibly recompose new ViT models with diverse scales. Thus, we discuss related works including model initialization and model compression which have similarities with the decomposition and recomposition of our method, respectively.

**Model Initialization.** Model initialization plays an important role in the training of deep neural networks, affecting both the rate of convergence and the quality of generalization. Fundamental methods, such as Xavier initialization [15] and Kaiming initialization [21], have been seminal in this domain. However, a recent trend is to use pre-trained models for initialization, as shown by studies such as [20, 5, 57, 32, 50]. This practice revolves around using these pre-trained models as a starting point, and then fine-tuning them for specific tasks. While such pre-trained models offer a superior initialization point, often outperforming the likes of Xavier and Kaiming initialization and ensuring faster training convergence, they come with their own challenges. Their large architectural footprint makes them unsuitable for direct deployment across various application scenarios, especially given the varying computing capabilities of devices. Furthermore, these pre-training methods often require

large amounts of data and computational resources, making it difficult to build specialized models for individual tasks.

**Model Compression**  Model compression [9, 55, 34] is a key area in deep learning, especially for deployment on resource-constrained devices. Micro-architecture requires the design of specific model architectures to meet different computational resource requirements, but the optimal architecture often varies depending on the required model size. Model pruning [64, 65] is a method that involves iteratively training the model, trimming less critical parameters, and subsequently fine-tuning for various model sizes. Quantization [18, 56, 14], which modulates the precision of model parameters,also requires unique iterative adjustments and potential retuning for each accuracy target. Meanwhile, knowledge distillation [33, 2], a concept popularized by [22], uses a more comprehensive teacher model to guide the training of a smaller student model. If we want to obtain $N$ student models, we need to train the model for $N$ times. A common drawback of these techniques is the lack of flexibility: generating $N$ models of different sizes typically requires $N$ times the computational and time cost.

**Prompt Tuning**  Visual Prompt Tuning (VPT) have also been proposed to improve parameter efficiency. For example, E^2VPT [17], which reduces tunable parameters through learnable prompts and pruning, and by works like Facing the Elephant in the Room [16] that examine optimal conditions for VPT based on task and data distribution. Methods such as AdapterFusion [41] and Prefix-Tuning [27] use small, task-specific modules or tunable prefixes to allow efficient model adaptation without full retraining. In multimodal tasks, Learnable In-Context Vector (L-ICV) [40] addresses in-context learning (ICL) challenges by improving VQA performance with reduced computational costs by distilling task information into a single learnable vector. While these approaches reduce the number of tunable parameters, they still lack flexibility in deployment. For instance, generating $N$ models of different sizes often requires $N$ separate tuning or retraining processes, leading to significant computational overhead.

Therefore, we introduce a novel training approach where we decompose the ViT model into different learngenes. These decomposed learngenes can be recomposed to generate models of diverse scales, making them adaptable to downstream tasks with different computational resource requirements. Additionally, by using these learngenes, the recomposed models also provide a good initialization. This results in improved performance and faster convergence in downstream tasks compared to learning from scratch.

## 3  Method

We propose to decompose a Vision Transformer (ViT) into a series basic learngenes that each layer of the ViT can be linearly recomposed by these learngenes. Figure 3 outlines the whole pipeline of our method, where Section 3.1 shows how to decompose the ViT into the basic learngenes and in Section 3.2, we detail how to get these learngenes during training. After decomposition, Section 3.3 demonstrates how to flexibly recompose these learngenes into diverse ViTs with different depths to achieve a the balance between parameter efficiency and model performance.

### 3.1  Decomposing the ViT

As discussed in Section 1, motivated by calculus, we propose to decompose a ViT into different basic learngenes where each layer of the ViT is the linear composition of these learngenes. Formally, given the basic learngenes $\boldsymbol{A}_1, ..., \boldsymbol{A}_N$, the parameters of the $l$-th layer $\boldsymbol{W}_l$ can be got by Equation 1. Note that $\boldsymbol{W}_l$ represents the general term of the parameters in the $l$-th layer of a ViT, which can be the modules such as the Multi-head Self Attention (MSA), Multi-layer Perceptron (MLP), and Layer Normalization (LN). In other words, each module of a ViT layer can be decomposed into a series of basic learngenes. For the coefficients $a(l, n)$, we assume that they satisfy a pre-defined polynomial such as the Taylor series, the first kind Chebyshev polynomial, or the Fourier series. Through preliminary experiments presented in Section 4.2.1, we find that the first kind Chebyshev polynomial is particularly well suited for our purposes. Therefore, we use it as an example for the decomposition coefficients in the following content. The first kind Chebyshev polynomial, represented by $T_n(x)$, is

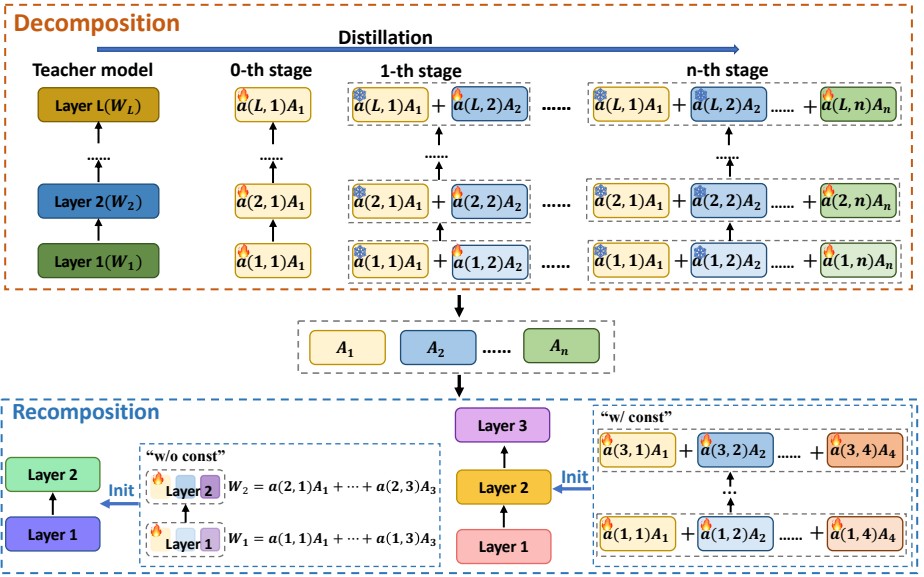

Figure 3: Decomposition and Recomposition. The upper part illustrates the decomposition process, where a ViT model is gradually recomposed into several learngenes. At each stage, only the newly added learngenes are trained, while the previously trained learngenes remain frozen. The lower part shows the recomposition process with two examples. The first initializes a 2-layer ViT with three learngenes trained by "without constraints", while the second initializes a 3-layer ViT with four learngenes trained by "with constraints". Note that the flame icons indicate that the parameters of the layer are trained, while the snowflake icons indicates that the parameters are frozen.

defined recursively as follows:

$$
\begin{aligned}
T_0(x) &= 1, \\
T_1(x) &= x, \\
T_{n+1}(x) &= 2xT_n(x) - T_{n-1}(x) \quad \text{for } n \geq 1.
\end{aligned}
\tag{2}
$$

Figure 2, it was observed that the parameters of each layer in the ViT have an approximately linear increasing trend. We assume that different layers of a ViT share the same learngene space, and the polynomial coefficients used to compose the learngenes are layer-dependent. This means that for $T_n = (x)$, where $x = \frac{l-1}{L}$, and the coefficient $a(l, n)$ is defined as follows:

$$
a(l, n) = T_n(\frac{l-1}{L}).
\tag{3}
$$

To avoid confusion, we use an example to show how to calculate these coefficients. Suppose the ViT model we decompose is a 12-layer ViT, in the other words, $L$ is equal to 12. For the second layer and the third order learngene, the formula to calculate $a(2, 3)$ is as follows:

$$
a(2, 3) = 2 \times \frac{2-1}{12} \times T_2(\frac{2-1}{12}) - T_1(\frac{2-1}{12}).
\tag{4}
$$

And for the fourth layer and the third-order learngene, the formula to calculate $a(4, 3)$ is

$$
a(4, 3) = 2 \times \frac{4-1}{12} \times T_2(\frac{4-1}{12}) - T_1(\frac{4-1}{12}).
\tag{5}
$$

Note that the first kind Chebyshev polynomial is recursive, so in each layer the coefficients of the subsequent learngenes are influenced by the coefficients of the previous learngenes.

## 3.2 Learning the Basic Learngenes

Here we introduce how to obtain the basic learngenes during training. Suppose we want to decompose a $L$-th layer ViT into $N$ learngenes. We iteratively use more learngenes, following Equation 1, to

construct and train the ViT. During training, in each iteration, only the newly added learngenes are trained, and all the previously used learngenes are fixed. More specifically, in the first iteration, we only use the first learngene $\boldsymbol{A}_1$ to build the ViT following Equation 1, i.e., $\boldsymbol{W}_l = a(l, 1)\boldsymbol{A}_1$, and train this ViT. After the first iteration, we add the second learngene $\boldsymbol{A}_2$ into the Equation 1, i.e., $\boldsymbol{W}_l = a(l, 1)\boldsymbol{A}_1 + a(l, 2)\boldsymbol{A}_2$, and then continue training this ViT on classification again. However, in this iteration, only the second learngene $\boldsymbol{A}_2$ will be trained, while $\boldsymbol{A}_1$ will be fixed. Then, we will iterate this process until all $N$ learngenes are trained in turn.

When training each learngene, we use the training objectives following MiniViT [62], where the total loss $\mathcal{L}$ contains two terms: the classification cross-entropy loss $\mathcal{L}_{ce}$ and the distillation loss $\mathcal{L}_{dl}$. The classification cross-entropy loss $\mathcal{L}_{ce}$ is as follows:

$$\mathcal{L}_{ce} = -\sum_j \sum_i y_{ij} \log(\phi_i(z_j)). \tag{6}$$

Here, $y_{ij}$ denotes the ground truth label for the $i$-th class of the $j$-th data instance, expressed in a one-hot encoded format. The function $\phi$ represents the softmax operation, and $z_j$ is the logit output corresponding to the $j$-th data instance from the model comprising with learngenes. Next, we define the distillation loss as follows:

$$\mathcal{L}_{dl} = KL(z_t || z_s), \tag{7}$$

where $z_t$ denotes the logits output of the teacher model, $z_s$ denotes the logits output of the model comprising with learngenes and $KL$ represents the Kullback-Leibler divergence loss.

Thus, the loss function for learngene-wise training integrates both cross entropy and distillation losses as follows:

$$\mathcal{L}_{all} = \mathcal{L}_{ce} + \mathcal{L}_{dl}. \tag{8}$$

### 3.3 Recomposing the ViTs

After the training process, the ViT model is decomposed into several learngenes. Consequently, these learngenes linearly composed by polynomial coefficients are serve as initialization for ViT models for diverse downstream tasks. It is important to note that, the linear coefficients are pre-defined by Equation 3, which means that these coefficients will not be updated during fine-tuning the recomposed models. The initialization of ViT models with layers of different depth use a suitable number of learngenes from Equation 3, with the number of layers $L$ adapted to fit the available computational resources. Based on the above decomposition formulas, the ViT models consisting of $L$ layers can be initialized as follows:

$$W_l = a(l, 1)A_1 + a(l, 2)A_2 + \ldots + a(l, k)A_k, \tag{9}$$

where $W_l$ denotes the parameters of the $l$-th layer in the ViT models, $k$ is the number of learngenes used for initialization.

After initializing the ViT models, there are two ways to train the model as shown in the lower part of Figure 3. The first one is that Equation 9 merely provides an initialization for the ViT models. In this way, the ViT models are trained without being constrained by Equation 9, the parameters of each layer $W_l$ are updated independently, i.e., the number of trainable parameters correlates with the layer number of the recomposed models. The second one is that the parameters of each layer of the ViT models are still constrained by Equation 9. In other words, during training, the ViT models update the parameters of the $k$ learngenes used for initialization, not the parameters of $W_l$. Consequently, the number of trainable parameters in the model depends on the number of learngenes used. Therefore, not only can we choose an appropriate number of learngenes to initialize models with an appropriate number of layers, but we can also choose different training methods based on computational resources as discussed in Section 4.2.3.

## 4 Experiments

In this section, we first describe the implementation details of the decomposition and recomposition processes of ViT in Section 4.1. Followed by experiments to validate that the ViT model can be decomposed into a series of learngenes and the feasibility of the ViT models that are flexibly composed of these learngenes in Section 4.2.

### 4.1 Implementation Details

#### 4.1.1 Datasets

To train each learngene, we use ImageNet-1K [12], which contains approximately 1.2M training images across 1000 classes and 50K validation images. After recomposing ViT models of different layers with learngenes, we adapt them on 9 diverse downstream datasets, which include 3 object classification tasks: CIFAR-10 [26], CIFAR-100 [26], and Tiny-ImageNet [1]; 5 fine-grained classification tasks: iNaturalist-2019 [66], Food-101 [4], Oxford Flowers-102 [35], Stanford Cars [25], and Oxford-IIIT Pets [38]; 1 texture classification task: DTD [10]. Each dataset presents unique challenges, ranging from basic object recognition to more specialized classification based on fine-grained visual differences and texture patterns.

#### 4.1.2 Details of Decomposition and Recomposition

**Decomposition.** Our approach to decompose the ViT model into different learngenes is based on the DeiT architecture [48]. To train the learngenes, we use a distillation strategy with the RegNet-16GF [42] as the teacher model. We train a total of 12 learngenes because there are 12 layers in DeiT. Constrained by computational resources, instead of training each learngene sequentially, we divided the training process into five phases, with learngene counts of $\{1, 2, 4, 8, 12\}$. For example, during the third training phase, only the two newly added learngenes are trained, while the two previously trained learngenes are frozen. In each phase, the model is trained for 100 epochs with an initial 5 warmup epochs.

**Recomposition.** In the recomposition, we can choose different number of learngenes to flexibly generate the ViT models with different layers, such as $\{2, 4, 6, 8, 10, 12\}$ layers. First, we determine the appropriate number of learngenes and number of layers of the ViT models based on computing resources. Then we use the pre-defined Equation 3 to calculate the polynomial coefficients. Following the Equation 9 can be used to initialize the parameters of the ViT models. After the models are initialized,they can be trained in one of two ways as mentioned in Section 4.2.3.

**Computing resources.** The resource cost of our method includes the decomposition and recomposition. In decomposition, it takes **500 epochs** to obtain 12 learngenes, while recomposition requires no additional training to use different number of decomposed learngenes to initialize different layer-based models. Since the number of layers of ViTs ranges from 1 to 12, and each layer can be initialized with 12 different numbers of learngenes, there are a total of $12 \times 12$ combinations of ViTs with different sizes and different initializations. Then if we try to get all these student ViTs by pre-training/distillation, we need to respectively train each one individually. Taking [62] as an example, training one student takes 300 epochs, then **$12 \times 12 \times 300 = 43.2K$ epochs** are required in total.

### 4.2 Results and Analyses

In this paper, we have two elementary assumptions of Equation 2. Firstly, a ViT can be decomposed into a series of basic learngenes. Secondly, ViTs with diverse depths can be flexibly recomposed by parts or all of these learngenes. Here, we first determine the polynomial coefficients for decomposition and recomposition based on the experiments in Section 4.2.1. Then we implement experiments to validate these two major assumptions in Section 4.2.2 and Section 4.2.3, respectively.

#### 4.2.1 Choosing Suitable Polynomial Coefficients

In the decomposition and recomposition, there are several choices of the polynomial coefficients for Equation 3, e.g., Taylor series, the first kind Chebyshev polynomial, Fourier series and Legendre series. To select the appropriate polynomial coefficients, we perform a simple experiment in which the ViT model is decomposed

Table 1: Results of ViTs with different decomposition coefficients.

|          | Taylor | Chebyshev | Fourier | Legendra |
| -------- | ------ | --------- | ------- | -------- |
| Accuracy | 74.80% | **77.73%** | 77.09%  | 73.13%   |

into 12 learngenes. We then use these polynomial coefficients to compose these learngenes and train them simultaneously for 100 epochs. The results are presented in Section 4.2.1, it can be observed that when using the first kind Chebyshev Polynomial coefficients, we have the highest performance:

77.73% accuracy. Therefore, we use the first kind Chebyshev polynomial coefficients in Equations 1 and 9 in the subsequent decomposition and recomposition experiments.

### 4.2.2 Decomposing the ViT

In Section 3.2, we show that the basic learn-genes decomposed from the ViT model, i.e., DeiT-Base, can be trained incrementally, a method named ICT. Figure 4(a) also shows the result of training all learngenes simultaneously under the same training settings, a method called SCT. The process of incremental training is visually represented in Figure 4, where it is evident that as the number of learn-genes increases, the accuracy of the ViT model composed of these learngenes not only improves, but also surpasses the results of SCT.

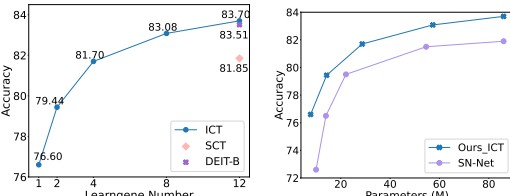

(a) Compare with Deit-Base  (b) Compare with SN-Net

Figure 4: The accuracy of our method is compared with that of DeiT-B and SN-Net on ImageNet-1K.

In particular, at 12 learngenes, the accuracy of the ViT model reaches 83.70%, which is comparable to the performance of DeiT-Base, where the parameters in different layers are initialized independently. The results shown by DeiT in Figure 4(a), which are the same as the training method used in the decomposition, are obtained by training with the loss function in Equation 8, with 500 epochs of training. Thus, after separately learning each linear learngene of the ViT, the performance can achieve the results of the classical ViT which is trained as a whole, confirming the earlier assumption in Section 1 that the parameters of ViT can be linearly decomposed.

Moreover, we also compare the decomposition results with SN-Net [37] in Figure 4(b), which describes the generation of differently scaled ViTs on ImageNet-1k. The results show that the models obtained by decomposition outperform those stitched by SN-Net from DeiT-Tiny, DeiT-Small, and DeiT-Base. The computational cost for both approaches is as follows: our method totals 800 epochs, consisting of 300 epochs for training DeiT-Base and 500 for the decomposition. In contrast, SN-Net incurs 1150 epochs, with each of DeiT-Tiny/Small/Base requiring $300 \times 3$ epochs, plus an additional $50 \times 5$ epochs for five stitched models. Thus, our method demonstrates not only better performance but also greater efficiency in computational cost.

### 4.2.3 Recomposing the ViTs

After training, these learngenes can be flexibly recomposed into ViT models with different layers according to Equation 9. To validate the effectiveness of recomposing ViTs with diverse depths by using different numbers of learngenes, we perform the following experiments.

**Evaluating Learngene Number and Layer Depth Effects on Model Performance.** As mentioned in Section 1, the decomposed learngenes can be used to initialize ViT models with different layers to satisfy the computational resource requirements. Additionally, Section 3.3 introduces two ways for training the reconstituted ViT models. The first method, named **"w/o const"** means that $W_l$ calculated by Equation 9 only provides an initialization, and the parameters of each layer are independent. The second method, named **"w/ const"** is that the parameters of each layer still satisfy Equation 9. Due to space limitations, some experimental results are presented in the Appendix.

In Figure 5, we have two dimensions: the depth of the network and the number of the learngenes used to recompose. From these figures, first, we can find that when recomposing a deeper ViT or using more basic learngenes to recompose ViTs, the performance will also improve in most cases, whether "w/ const" or "w/o const" are used. In addition, increasing the number of layers tends to have a greater impact on the preform of the models. Second, on each graph, there is a boxed area bounded by red contours that represents 95% of the maximum value on each graph. Therefore, if the computer resources of clients are limited, they can choose to build VITs with fewer layers and initialize with fewer learngenes to achieve better resource-performance trade-off. For example, in Figure 5(g), the 6-layer ViT model initialized by only 1 learngene can achieve 95% of the best performance.

Figures 5(f) and 5(g) also show a comparison between the "w/ const" and "w/o const" training methods. However, it should be noted that when the number of layers exceeds the number of learngenes, the model trained by the "w/o const" method consumes more memory than the "w/

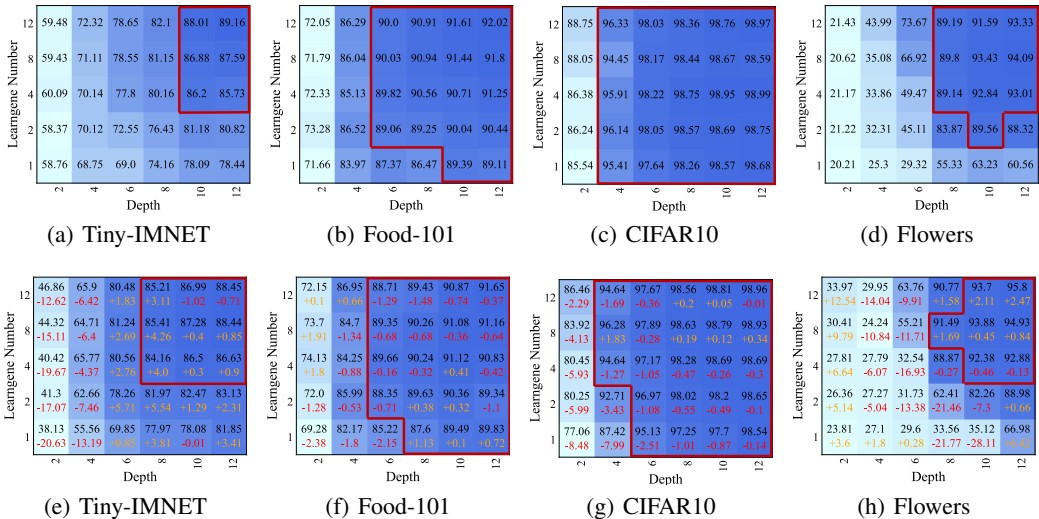

Figure 5: Performance of ViTs with different layers initialized with different learngenes on downstream datasets. The first row shows the results with the "w/o const" training method, the second row is trained with "w/ const". The second row also shows the differences between the two, where "+" indicates improvement and "−" indicates degradation. The boxed areas bounded by the red contours represent regions where the accuracy is within 95% of the maximum value of each graph.

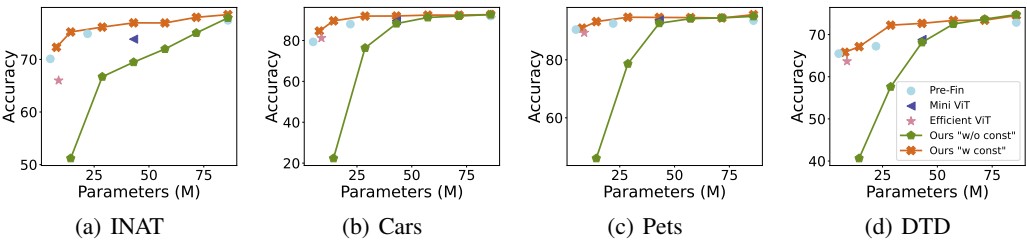

Figure 6: Comparative results. The ViTs are initialized with two different recomposition cases and trained by corresponding methods, i.e., one is trained with constraint and the other is trained without constraint, and then compared with the performance of differently scaled pre-trained models and models obtained by model compression methods.

const". Therefore, for downstream clients with limited computational resources who want better performance. They can construct a ViT model with a higher number of layers, initialize it with a smaller number of learngenes, and then use "w/ const" training method. For example, a 12-layer ViT model initialized with two learngenes requires 86MB of memory using the "w/o const" method, but only 14.3MB using the "w/ const" method. On Food-101, it achieves 89.34% performance with "w/ const" and 90.44% with "w/o const". Despite the slightly lower performance, the "w/ const" method significantly reduces memory usage by about 12 times.

**Comparative Analysis.** We compare our method with pre-training & fine-tuning (Pre-Fin) and model compression methods. In particular, (1) **Pre-Fin**: We use the pre-trained DeiT models including DeiT-Tiny, DeiT-Small and DeiT-Base in [48], and then fine-tune the model for downstream tasks. (2) **Model compression:** The methods used include Mini-DeiT-B in [62] and Efficient-ViT [30].

For two training strategies in recomposition, we use different learngene configurations to recompose models with different numbers of trainable parameters. For the "w/ const", the parameters of the model depend on the number of learngenes as in Section 3.3. Therefore, we use the 12-layer ViT model recomposed by {1, 2, 4, 6, 8, 10, 12} learngenes. For the "w/o const", the trainable parameters correlate with the layer number of the model. Here, the number of learngenes is kept at 12 and ViT models are initialized over {2, 4, 6, 8, 10, 12} layers. Additional dataset results and corresponding numerical data are in the Appendix due to space limitations.

From Figure 6, we find that when different trainable parameters are used, "w/ const" outperforms the other methods such as pre-fine and model compression on downstream datasets. For example, in

Figure 6(a), when the model with 7.8M parameters is recomposed from a single learngene and trained with constraints, the accuracy is 72.3%. This performance exceeds that of Efficient ViT, which has 8.8M parameters, by 6.3%. Moreover, "w/o const" performs slightly worse than other methods with less trainable parameters because the networks have fewer layers. As in Figure 6(b), a 4-layer model with 28M parameters trained with "w/o const" performs over 9% worse than DeiT-Small with 22M parameters. However, as the number of layers increases, the performance of models trained with "w/o const" improves. For example, a 6-layer model with the same number of parameters as MiniViT, which is a 12-layer model, shows a performance difference of only 1.4%, indicating competitive results. These comparisons validate that when using suitable training strategy, the recomposed diverse-scale ViTs have good initialization that by simply training, they can achieve comparable or better performance than Pre-Fin and model compression methods. Furthermore, by our strategy, an appropriate number of decomposed learngenes can be selected based on available computational resources to initialize the models without training from scratch, achieving flexible accuracy and efficiency trade-offs.

## 5    Conclusion

In summary, this paper proposes a novel training method for efficiently generating ViTs of varied sizes to meet diverse computational needs. By employing linear decomposition, a ViT can be decomposed into basic learngenes, termed as "learngenes", which encapsulate critical knowledge from the original model. These learngenes can then be selectively and linearly recomposed to form ViTs of various layers and sizes. This flexible recomposition provides an economical and adaptive solution for creating a series of small or medium ViTs tailored to different deployment environments. Our experiments confirm the effectiveness of this learngene-based decomposition-recomposition method and show that these recomposed ViTs maintain performance comparable to traditional model compression techniques while offering greater flexibility and efficiency.

## Acknowledgments and Disclosure of Funding

This research was supported by the National Science Foundation of China (62125602, 62076063), the Key Program of Jiangsu Science Foundation (BK20243012), the Fundamental Research Funds for the Central Universities (2242024k30035) and the Big Data Computing Center of Southeast University.

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

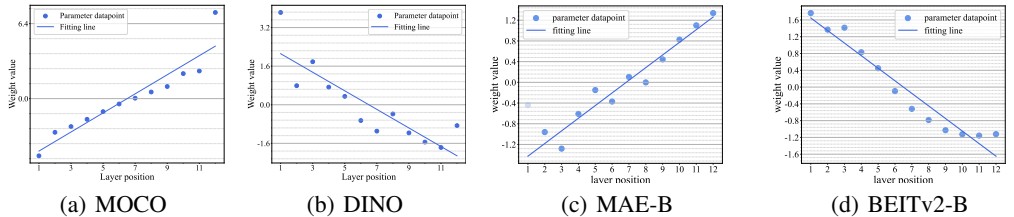

| (a) MOCO | (b) DINO | (c) MAE-B | (d) BEITv2-B |

Figure 7: The parameters of each layer of the pre-trained model and their corresponding layer position relationships.

# A   Appendix / supplemental material

In the following section, we present the process of dimensionality reduction applied to the parameters of each layer in the Vision Transformer (ViT) models in order to observe the interrelationships between the parameters across different layers. We then supplement the experimental content presented in the main text, including the experimental results on decomposition and recomposition, as well as implementation details. Finally, this section concludes with a discussion of the limitations and societal implications of our method.

## A.1   Dimensionality Reduction and Inter-Layer Parameter Relationships in ViTs

Our analysis starts with the aggregation of parameter matrices from the Multihead Self-Attention (MSA), Multilayer Perceptron (MLP), and Layer Normalization (LN) modules of an extensively trained Transformer model. Specifically, within the MSA module and given an input $x$, we first perform the computation of $(xW^Q)(xW^K)^T$ to determine the attentional outputs for each head. This then facilitates the derivation of $W^A$ for the MSA as follows:

$$W^A = W^Q W^{K^T} \tag{10}$$

where $W^K$ and $W^Q$, belonging to $\mathbb{R}^{D \times D}$, act as transformation parameter matrices for keys and queries, respectively, where $D$ denotes the dimensionality of the intermediate embeddings. Subsequent operations involve the bias vectors for keys and queries, proceeding according to Equation 10. Together with other parameter matrices in the MSA module, we reshape these into vectors and subject them to L2 normalization.

In the MLP module, we flatten and normalize weight matrices and bias vectors from two linear transformations. A similar approach is employed in the LN module, where weight matrices and bias vectors associated with transformation parameters are normalized. Normalized vectors, specific to each parameter type, are concatenated in alignment with their corresponding layer, culminating in a synthesized matrix $S \in \mathbb{R}^{N \times D_L}$, where $D_L$ symbolizes the normalized vector dimension for each parameter type, and $N$ signifies the depth of the Transformer.

Principal Component Analysis (PCA) [24] is then utilized to reduce the dimension of each vector in $S$ to a one-dimensional scalar, primarily for analytical convenience. These reduced vectors are then combined to form the matrix $H \in \mathbb{R}^{N \times K}$, with $K$ indicative of the number of parameter types. Each row of $H$ is interpreted as the representative vector of the corresponding layer. Another iteration of PCA is performed to compress the rows of $H$ into a one-dimensional domain, resulting in a vector $U \in \mathbb{R}^N$. This final vector is used to illustrate the relationship between the sequential positioning of the layers in the transformer and their respective parameter values, thus illustrating the nuanced interplay of parameters across the architectural layers.

We also show the graphs for MoCov3 [8], DINOv2 [36], MAE-B [19] and BEITv2-B [3] in Figures 7(a) to 7(d), which show that the parameters of the layers have an approximately linear correlation with the layer position.

## A.2   Experiment

### A.2.1   Decomposition

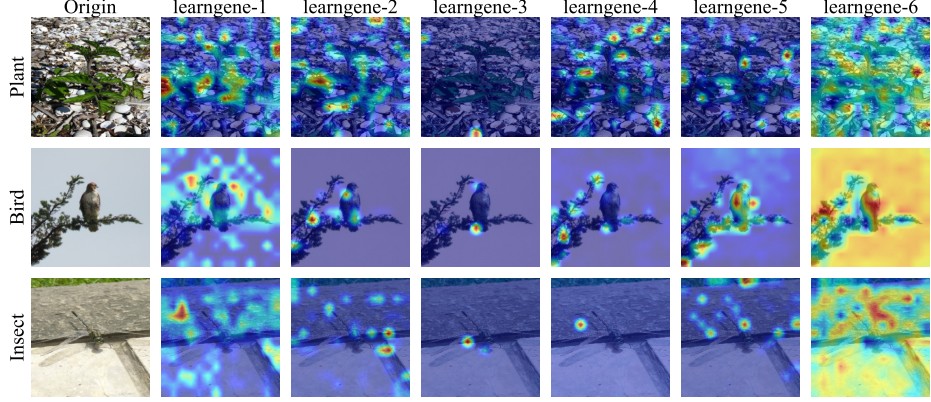

(a) Qualitative Visualization of Each learngene. Uses each of the first six learngenes to initialize the model separately, showing different characteristics and areas of focus for each learngene.

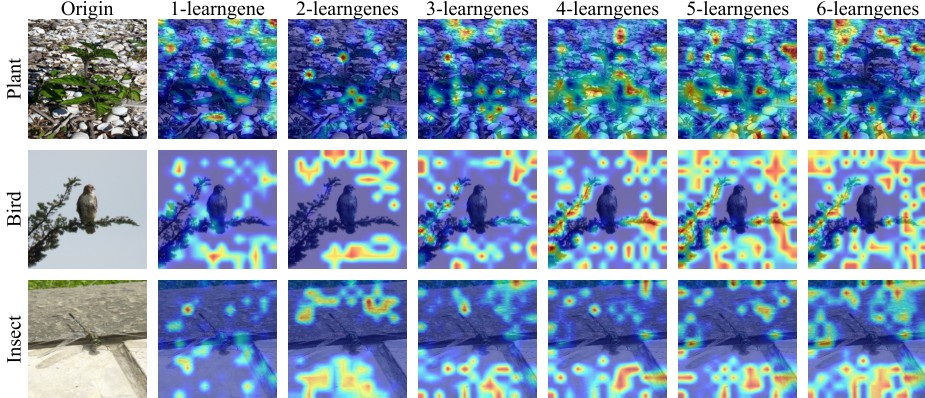

(b) Qualitative visualization of the ViT model initialized by incremental learngenes.

Figure 8: Qualitative visualization of decomposed learngenes

**Qualitative Visualization of Learngenes.** After training, we obtain the decomposed learngenes of the Vision Transformer (ViT). To analyze their characteristics, we employ Gradient-weighted Class Activation Mapping (Grad-CAM) [44], which is pivotal in the classification process, to visualize the feature maps in two scenarios: one where each learngene is used independently to initialize a ViT, and the other where learngenes are added incrementally to initialize the ViT model.

From Figure 8(a), we find that each learngene focuses on uniquely specific regions, indicating that the functionalities of the ViT model are decomposed into distinct learngenes. Specifically, the first learngene focuses on basic shapes and contours. The second learngene detects textures and colors, such as highlighting leaf textures and bird feather details. The third learngene identifies spatial features such as insect wings, while the fourth learngene detects finer details such as bird eyes or insect antennae. The fifth learngene emphasizes image contrast, and the sixth distinguishes between foreground and background, focusing on depth perception. As shown in Figure 8(b), early learngenes capture basic shapes and edges, while later learngenes refine the focus on textures, colors, and specific details such as leaf veins or bird feathers. As learngenes are added, the attention of model shifts to finer features such as bird beaks, insect wings, and antennae, indicating a progression from general to detailed feature recognition that is likely to lead to more accurate image classification.

### A.2.2 Recomposition

**Evaluating Learngene Number and Layer Depth Effects on Model Performance.** The experimental results in Figure 9 on these two datasets further verify the findings in the text. The results in the appendix confirm the observation that deeper network configurations or a larger number of

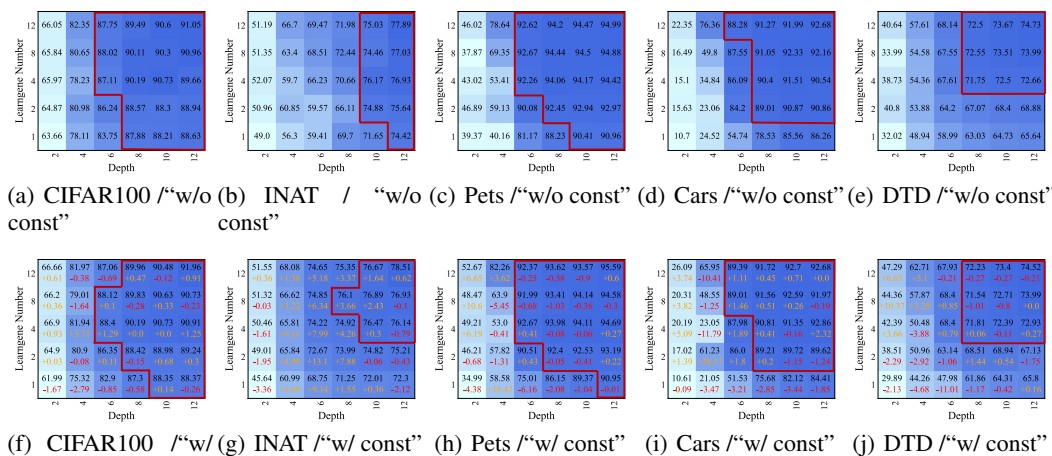

(a) CIFAR100 /"w/o const"
(b) INAT / "w/o const"
(c) Pets /"w/o const"
(d) Cars /"w/o const"
(e) DTD /"w/o const"

(f) CIFAR100 /"w/ const"
(g) INAT /"w/ const"
(h) Pets /"w/ const"
(i) Cars /"w/ const"
(j) DTD /"w/ const"

Figure 9: Performance of ViTs with different layers initialized with different learngenes on downstream datasets.

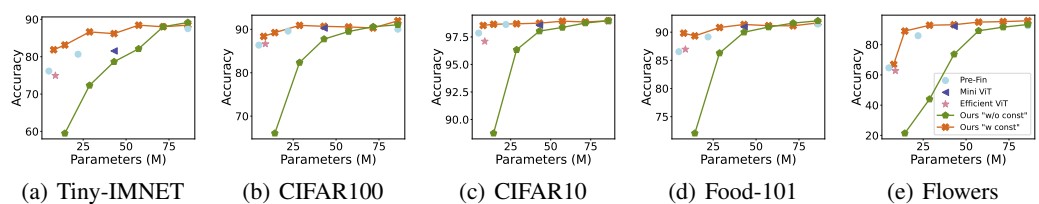

(a) Tiny-IMNET
(b) CIFAR100
(c) CIFAR10
(d) Food-101
(e) Flowers

Figure 10: Comparative results. The ViTs are initialized with two different recomposition cases and trained by corresponding methods, i.e., one is trained with constraint and the other is trained without constraint, and then compared with the performance of differently scaled pre-trained models and models obtained by model compression methods.

decomposed learngenes generally improve ViT performance, whether trained by "w/o const" or "w/ const". The results also show stable performance beyond certain learngene and layer thresholds, and provide strategies for optimizing ViT models under resource constraints, such as using fewer layers and learngenes to achieve near-optimal performance. We also compare the "w/const" and "w/o const" training methods and find that performance differences decrease as model complexity increases. In particular, models with higher layers show minimal performance variance between the two methods, but differ in memory consumption, with "w/o const" requiring more. We recommend "w/o const" for efficiency in resource-constrained scenarios, as shown by the 12-layer ViT model initialized with fewer learngenes, which achieves comparable performance with significantly less memory consumption.

**Comparative Results.** The experimental results presented in Figure 10 show that a 12-layer Vision Transformer (ViT) initialized with different numbers of learngenes and trained using the "with constraint" method, represented by the orange lines in the graphs, outperforms traditional approaches such as pre-fine tuning and model compression under the same parameter constraints. In addition, as the number of layers in the model increases to a certain threshold, models initialized with 12 learngenes and trained using the "without constraint" method also outperform these baselines. Thus, our method strikes a balance between flexibility and model efficiency.

**Numerical Results for Comparative Results.** Table 2 provide the numerical performance on 9 downstream tasks to compare our methods with pre-training fine-tuning and model compression methods. The same results are also shown in Figures 6 and 10 of the main paper.

Table 2: Comparative experiments. Two cases of initializing models with learngenes and their corresponding training approaches, compared with the pre-training fine-tuning and model compression methods.

| Model | Para | CIFAR10 | CIFAR100 | Tiny-ImageNet | INAT | Food | Cars | Flowers | Pets | DTD |
|---|---|---|---|---|---|---|---|---|---|---|
| | | | | Model Pretrained | | | | | | |
| Deit-T | 5M | 97.85 | 86.36 | 76.15 | 70.1 | 86.54 | 79.32 | 64.69 | 90.43 | 65.48 |
| Deit-S | 22M | 98.63 | 89.59 | 80.66 | 74.88 | 89.16 | 88.01 | 85.92 | 92.53 | 67.23 |
| Deit-B | 86M | 98.92 | 90.03 | 87.5 | 77.43 | 91.44 | 91.97 | 92.54 | 93.54 | 72.87 |
| | | | | Mini-Deit and Efficient-ViT | | | | | | |
| Mini-Deit | 43M | 98.57 | 90.29 | 81.54 | 73.85 | 90.94 | 89.68 | 92.15 | 93.71 | 68.83 |
| Efficient-ViT | 8.8M | 97.1 | 86.63 | 74.95 | 66 | 86.96 | 81.2 | 62.75 | 89.35 | 63.67 |
| | | | 12 learngenes used to recompose different layers ViTs trained with "w/o const" training method | | | | | | | |
| 2layer-12learngene | 14.3M | 88.75 | 66.05 | 59.48 | 51.19 | 72.05 | 22.35 | 21.43 | 46.02 | 40.64 |
| 4layer-12learngene | 28.7M | 96.33 | 82.35 | 72.32 | 66.7 | 86.29 | 76.36 | 43.99 | 78.64 | 57.61 |
| 6layer-12learngene | 43M | 98.03 | 87.75 | 78.65 | 69.47 | 90 | 88.28 | 73.67 | 92.62 | 68.14 |
| 8layer-12learngene | 57.3M | 98.36 | 89.49 | 82.1 | 71.98 | 90.91 | 91.27 | 89.19 | 94.2 | 72.5 |
| 10layer-12learngene | 71.6M | 98.76 | 90.6 | 88.01 | 75.03 | 91.61 | 91.99 | 91.59 | 94.47 | 73.67 |
| 12layer-12learngene | 86M | 98.97 | 91.05 | 89.16 | 77.89 | 92.02 | 92.68 | 93.33 | 94.99 | 74.73 |
| | | | 12 layered models recomposed by different number of learngenes, trained with "w const" training method | | | | | | | |
| 12layer-1learngene | 7.8M | 98.54 | 88.37 | 81.85 | 72.3 | 89.83 | 84.61 | 66.98 | 90.95 | 65.8 |
| 12layer-2learngene | 14.3M | 98.65 | 89.24 | 83.13 | 75.21 | 89.34 | 89.62 | 88.98 | 93.19 | 67.13 |
| 12layer-4learngene | 28.7M | 98.69 | 90.91 | 86.63 | 76.14 | 90.83 | 91.95 | 92.88 | 94.69 | 72.23 |
| 12layer-6learngene | 43M | 98.75 | 90.69 | 86.18 | 76.95 | 91.37 | 91.99 | 93.2 | 94.63 | 72.66 |
| 12layer-8learngene | 57.3M | 98.93 | 90.73 | 88.44 | 76.93 | 91.16 | 92.43 | 94.93 | 94.58 | 73.35 |
| 12layer-10learngene | 71.6M | 98.86 | 90.31 | 88.01 | 78 | 91.13 | 92.42 | 95.24 | 94.44 | 73.4 |
| 12layer-12learngene | 86M | 98.96 | 91.96 | 88.45 | 78.51 | 91.65 | 92.68 | 95.8 | 95.59 | 74.52 |

Table 3: Overview of Classification Datasets

| Dataset Name | Categories | Training Samples | Test Samples |
|---|---|---|---|
| CIFAR-10 | 10 | 50,000 | 10,000 |
| CIFAR-100 | 100 | 50,000 | 10,000 |
| Tiny-ImageNet | 200 | 100,000 | 10,000 |
| INAT-2019 | 1,010 | 268,243 | 64,401 |
| Food-101 | 101 | 75,750 | 25,250 |
| Stanford Cars | 196 | 8,144 | 8,041 |
| Oxford Flowers | 102 | 2,040 | 818 |
| Oxford-IIIT Pets | 37 | 3,680 | 3,669 |
| DTD | 47 | 3,760 | 1,880 |

## A.3 Implementation Details

### A.3.1 Code

We implement the model using PyTorch [39] and the Timm library [53]. The decomposition process is trained over 500 epochs on four NVIDIA RTX 3090 GPUs, and the recomposed models are trained over 100 epochs on two NVIDIA RTX 3090 GPUs for each downstream task.

### A.3.2 Datasets and Pre-processing

**Datasets** We adapt the recomposed models on 9 diverse downstream datasets, covering a wide range of classification challenges. These datasets include 3 object classification tasks: CIFAR-10 [26], CIFAR-100 [26], and Tiny-ImageNet [1]; 5 fine-grained classification tasks: iNaturalist-2019 [66], Food-101 [4], Oxford Flowers-102 [35], Stanford Cars [25], and Oxford-IIIT Pets [38]; 1 texture classification task: DTD [10]. The details of these datasets are in Table 3.

**Data Process** Following previous works [62], we train and evaluate the decomposition and recomposition processes on all datasets at a resolution of $224 \times 224$. The data augmentation techniques employed include RandAugment [11], Cutmix [60], Mixup [61], and random erasing.

### A.3.3 Hyper-parameter

**Decomposition** We train each learngene using the following setting:

- Batch Size: We employ distributed training across 4 GPUs, with each GPU handling 128 data instances, resulting in an overall batch size of 512.
- Optimizer: The training of each learngene is optimized using AdamW, with an initial learning rate of 0.0008 and a weight decay of 0.05.
- Learning Rate Schedule: We apply a cosine learning rate decay, with a warm-up period of 5 epochs.

**Recomposition**    We train the recomposed models on downstream tasks using the following setting:

- Batch Size: We employ distributed training across 2 GPUs, with each GPU handling 128 data instances, resulting in an overall batch size of 256.
- Optimizer: The training of each learngene is optimized using AdamW and a weight decay of 0.05.
- Learning Rate Schedule: We apply a cosine learning rate decay, with a warm-up period of 5 epochs.

