# OpenReview forum: "Linearly Decomposing and Recomposing Vision Transformers for Diverse-Scale Models"
_NeurIPS.cc/2024/Conference — NeurIPS 2024 poster_

### Official Review · Reviewer_caYK · 2024-07-03

**Soundness:** 3
**Presentation:** 2
**Contribution:** 3
**Rating:** 6
**Confidence:** 5

**Summary:**

This paper presents a novel approach to adapting Vision Transformer (ViT) models for diverse computational environments. The key idea is to linearly decompose a large ViT model into a set of basic components during training, which can then be flexibly recomposed to create smaller, pre-initialized models suitable for various deployment scenarios.

**Strengths:**

The work makes three key contributions: 1. The paper introduces the idea of linearly decomposing ViT layers into shared components, which is a novel approach in the field of model adaptation. 2. Unlike traditional compression methods that require separate training for each model size, this approach allows for flexible model generation from a single decomposition. 3. The flexibility in recomposing models of various scales addresses the challenge of deploying ViTs in environments with different computational constraints.

**Weaknesses:**

There are several concerns regarding this work:

1. The training process for the decomposed modules requires a more detailed description.

2. The implementation details of the reconstruction process are insufficiently explained.

3. The empirical design does not fully support the proposed method. Many tunable components are not thoroughly discussed, such as the training and combination strategies for each layer's sub-modules, and the achieved latency savings at different scales.

4. The literature review is superficial. Parameter-efficient paradigms should be comprehensively discussed to clearly differentiate the proposed solution from existing approaches.

5. The work appears to be at an early stage of development. While the motivation and design show promise, the writing and formatting require significant improvement. The authors should enhance the overall writing quality and correct numerous typographical errors throughout the manuscript (e.g., in the caption of Figure 2, "layer poaition" is misspelled, and there should be a space before parentheses).

**Questions:**

1. How to train each of the sub-components so the cumulative performance can be on par with the original model? Due to the parameter gap, given the same amount of data, smaller modules may not perform equivalently to the bigger model. With that said, a model of Wn may perform better than the cumulative power of {A1...An}. Did the authors make any effort to address this challenge?

2. I have trouble understanding this statement: "Interestingly, as shown in Figure 2 of [39], the parameters of most layers in the well-trained ViT of [10] have an approximately non-decreasing trend after PCA dimension reduction." What do the x and y axes represent? What is the figure telling us?

3. In the empirical design, where can I see your arrangement of sub-module combinations {A1...An}? From lines 222 to 225, I only see layer-level combinations. Also, the empirical results should include the latency improvement for each scale as it is an important motivation of this paper.

3. I have trouble understanding this statement: "For example, during the third training phase, only the two newly added components are trained, while the two previously trained components are frozen. In each phase, the model is trained for 100 epochs with an initial 5 warmup epochs." Shouldn't this be one added component, not two?

**Limitations:**

Please refer to weakness and questions.

---

> ### Author Rebuttal · Authors · 2024-08-07
>
> To W1 \& Q1: Details of decomposition
>
> Thank you for your question. Due to the parameter gap, the 12-layer model with only one component $A_1$ in the first stage does not perform as well as a typical 12-layer model. However, as more components are trained, the performance becomes comparable to the model with the same parameters.
>
> As shown in Fig.C of the PDF, using DeiT-Base as an example, ${W_l}$ are the layer parameters of the model and are constructed by trainable parameters ${A_n}$ (include all modules in a ViT layer like MSA, MLP). In the 1-st stage, each layer $W_l$ initially contains one component, $A_1$, shared across all layers, with coefficients calculated using a predefined polynomial (Eq.3). The number of learnable parameters in the 12-layer model is approximately equal to $A_1$, achieving 76.6% performance, as in Fig.4(a).
>
> In the 2-nd stage, a new component, $A_2$, is added to each layer, and $A_1$ and $A_2$ are combined using their respective coefficients to initialize each layer for forward propagation as Eq.1. During training, the parameters of $A_1$ remain fixed, and only the newly added component $A_2$ is updated. In this way, the learnable parameters still equals to the parameters of one component. This process is repeated until all components are fully trained.
>
> As the number of added components increases, the performance of the 12-layer model improves, reaching 83.70\% with 12 components, which is comparable to DeiT-Base trained as a whole, with a performance of 83.51\% under the same training conditions and parameters. This confirms our assumption that each ViT layer can be linearly decomposed into a shared component {$\{A_1, \ldots, A_n\}$}, and that a model composed of components {$\{A_1, \ldots, A_k\}$} can perform comparably to the full model $W$ under the same parameters as in Fig.6. We will include these details in the revision for clarity. Should you have any further suggestions, we would be happy to address them.
>
> To W2: Details of recomposition
>
> After all components are trained,  the components can be directly used to initialize model without training in the recomposition. In Fig.D of the PDF, we present two examples. Regardless of the training method, different model structures are initialized by Eq.9. Under the "without constraint" training method, Eq.9 is only used to initialize $\{W_l\}$; the trainable parameters of each layer ($\{W_l\}$) are updated independently during training, the same as the normal ViT fine-tuning. The "with constraint" training method decomposed the $\{W_l\}$ to trainable parameters $\{A_k\}$ while keeping the polynomial coefficients fixed using Eq.9. This means that, during the gradient back-propagation, the gradient is backed to $\{A_k\}$ instead of $\{W_l\}$. We hope this explanation clarifies the reconstruction, and we will include these details in the revision.
>
> To W3 \& Q3:  Empirical design of components \& latency improvement
>
> Thank you for your questions. In the decomposition and recomposition, all sub-modules within a ViT layer are treated uniformly. Therefore, in our paper, each component $A_n$ includes all sub-modules within a ViT layer $W$, such as MSA, MLP, FFN, and others. As in Fig.C of PDF, in the different stages of decomposition, the parameters of each layer are obtained by Eq.1. In the recomposition, as in Fig.D, ViT models of varying layer can be initialized with different numbers of components, with each layer’s parameters initialized by Eq.9.
>
> Regarding the latency improvements for different scales, in recomposition, models of different sizes can be initialized with components by Eq.9, without the need for repeated training as required in knowledge distillation, which contributes to our latency improvement. In Sec.4.1.2 "Computing Resources", we compare the computational savings of our method against other model compression methods, highlighting the efficiency of our approach.
>
> To W4: Literature review
>
> We appreciate the suggestion to expand the literature review. Currently our literature review includes model initialization and model compression (Micro-architecture design, Model pruning, Quantization, and Knowledge distillation). In the revision, we will also provide a detailed discussion of other parameter-efficient paradigms, including Low-Rank Factorization (LRF) and Parameter Sharing (PS). For LRK and PS, generating models of different sizes typically requires redesigning the factorization or sharing scheme for each target model size, which can be complex and resource-intensive. In contrast, our approach allows flexible and economical generation of models with different sizes without repeated training or redesign.
>
> To W5: Writing
>
> Thank you for the feedback. We'll improve the revision by enhancing writing quality, correcting typos, and refining formatting for a more polished presentation.
>
> To Q2: Details of Fig.2
>
> Thank you for your question. In Fig.2, the x-axis is the layer index of ViT, while the y-axis is the values of the parameters in each layer after performing PCA, reduced to one dimension (details are in the Suppl Material Sec.A.1). The figure shows a linear relationship between these two variables. This observation suggests that the parameters across layers in a well-trained transformer are not independent variables, but may be linearly related in a higher dimensional space, possibly correlated with the layer index. This implies that the parameters can be reconstructed using the form given in Eq.1, enabling flexible initialization for models of different scales. We will clarify this point in the revision to enhance understanding.
>
> To Q4: Experimental design of decomposition training
>
> Thank you. You've raised an important point; ideally, components would be added and trained one at a time, but that's time consuming. To balance efficiency and performance, we trained multiple components in stages. We will clarify this point in the revision to ensure better understanding.

---

> > ### Comment · Reviewer_caYK · 2024-08-07
> >
> > Thank you for the detailed response. In general, this work can be categorized as parameter-efficient learning. With that said, the authors may also benefit from discussing the two seminal works [1][2].
> >
> > [1] E^ 2VPT: An Effective and Efficient Approach for Visual Prompt Tuning;
> > [2] Facing the Elephant in the Room: Visual Prompt Tuning or Full Finetuning?

---

> ### Author Response · Authors · 2024-08-08
>
> Thank you for your insightful comments and for pointing out the relevance of the two seminal works. We agree that these works are indeed crucial in the context of parameter-efficient learning. E^2VPT introduces a method for efficient fine-tuning of large-scale transformer models through learnable prompts and pruning techniques, reducing the number of tunable parameters while maintaining performance. Facing the Elephant in the Room analyzes when and why Visual Prompt Tuning (VPT) is effective, offering insights into its applicability depending on task objectives and data distributions. Both are closely aligned with our approach to parameter-efficient learning, and we will include discussions of these works in our revision to provide a more comprehensive overview of existing approaches in the field of parameter-efficient learning.

---

> > ### Comment · Reviewer_caYK · 2024-08-08
> >
> > Glad to hear our conversation has been helpful. I've completed the final rating—good luck!

---

> > > ### Author Response · Authors · 2024-08-09
> > >
> > > Thank you for your thoughtful feedback and support throughout the review process. Your insights are invaluable in helping us improve our work. We appreciate your time and effort, and we're grateful for the final rating.

---

### Official Review · Reviewer_GH5B · 2024-07-10

**Soundness:** 3
**Presentation:** 3
**Contribution:** 3
**Rating:** 7
**Confidence:** 4

**Summary:**

This paper proposes a novel approach to generate diverse-scale Vision Transformer (ViT) models with varying computational complexity, aiming to address the deployment challenges posed by the fixed architectures of standard ViT models. The key idea involves linearly decomposing a pre-trained ViT model into basic components and then recomposing these components to initialize ViT models of different scales, effectively enabling the economical generation of models tailored to diverse computational resource constraints.

**Strengths:**

The paper is well-organized and logically structured, with the introduction, methodology, and experiments seamlessly connected, making it easy to understand and evaluate.

The proposed decomposition-recomposition strategy is interesting which offers two key advantages: (1) an iterative training mechanism that generates a series of pre-trained models with different scales during the decomposition process, and (2) the flexibility to recompose the decomposed components into diverse-scale ViTs without additional training, enabling dynamic accuracy-efficiency trade-offs.

The provision of code in the supplementary material facilitates reproducibility of the experiments.

**Weaknesses:**

The paper could explain the component sharing approach in more detail. Is it similar to MiniViT, where components are shared across layers? Or are the component parameters only shared within each layer? Also, please clarify if the forward process is the same as in traditional ViTs.

For the recomposition, could you provide more information on the initialization methods used for the 'with constraint' and 'without constraint' training approaches? Are the initialization methods the same, with the only difference being the parameters being optimized (layer parameters W vs. component parameters θ)?

**Questions:**

Please see the Weaknesses.

**Limitations:**

Please refer to the Weaknesses.

---

> ### Author Rebuttal · Authors · 2024-08-07
>
> To 1: Components sharing approach
>
> Thank you for your insightful question. The component sharing in our model is applied within each individual layer, unlike the cross-layer sharing seen in MiniViT. Additionally, the forward process follows the typical ViT architecture, such as DeiT, during training. In the revision, we will include more detailed explanations to clarify these points.
>
> To 2: Training approaches in recomposition
>
> Thank you for your question. For the two training approaches, the initialization methods used are the same. The key difference lies in how the parameters are optimized during training. In the "without constraint" approach, Eq.9 is only used to provide an initialization of layer parameters $W_l$ for ViTs. During training, the parameters of each layer $W_l$ are then updated independently, the same as the normal ViT fine-tuning, without being constrained by Eq.9. In contrast, the "with constraint" approach optimizes the component parameters directly, rather than the individual layer parameters $W_l$. This means that during gradient back-propagation, the gradient is applied to ${A_k}$ instead of ${W_l}$. Therefore, the parameters of each layer remain constrained by Eq.9, with components shared across all layers. We will clarify this further in the revision.

---

### Official Review · Reviewer_DK4A · 2024-07-13

**Soundness:** 3
**Presentation:** 3
**Contribution:** 3
**Rating:** 7
**Confidence:** 4

**Summary:**

This paper proposes a novel method to efficiently adapt Vision Transformer (ViT) models to devices with diverse computational resources by linearly decomposing a ViT into basic components that can be flexibly recomposed into ViTs of various scales. The key ideas are:

- Inspired by polynomial decomposition, a ViT can be decomposed into basic components where each layer is a linear combination of the components using polynomial coefficients.
- The components are trained incrementally, where in each iteration only the newly added components are trained while previously trained ones are fixed. This element-wise training reduces compute cost.
- The trained components can then be flexibly recomposed into ViTs of different depths using an appropriate number of components to achieve accuracy-efficiency trade-offs for different deployment scenarios.
- Compared to repeatedly training diverse-scale models from scratch, this decomposition-recomposition strategy is more economical.

Experiments demonstrate the effectiveness of the method - the decomposed components can recompose ViTs of various scales that achieve comparable or better performance than traditional model compression methods.

**Strengths:**

1. The linear decomposition-recomposition idea, inspired by mathematical concepts, provides an elegant and principled framework to adapt ViTs to diverse computational constraints. The assumption that ViT layers can be decomposed into and recomposed from basic components is well-motivated and validated.
2. The incremental component-wise training is a smart strategy to reduce training cost while enabling the components to capture essential functionalities of the ViT. Limiting training to only the newly added components in each phase is efficient.
3. The flexible recomposition allows generating a spectrum of ViT scales by varying the number of layers and components used. This provides a smooth accuracy-efficiency trade-off to cater to different deployment requirements, which is a key practical advantage.
4. Extensive and rigorous experiments convincingly demonstrate the effectiveness of the method. The decomposed models perform competitively or better compared to strong compression baselines. Qualitative visualizations provide good insight into the progressive feature learning by the components.
5. Overall, the framework attacks an important problem and proposes an innovative, principled, and effective solution. The writing is mostly clear and easy to follow.

**Weaknesses:**

I have three major concerns, I will consider adjusting the score based on the quality of the rebuttal provided.

1. the recomposition process can be explained more clearly, especially the two types of coefficient initialization and the two training strategies ("with constraint" and "without constraint"). A figure illustrating the different recomposed architectures will help understanding. Regarding the statement in line 198 that the 'without constraint' training method still adheres to the constraints of Equation 9, could you clarify if this means the polynomial coefficients remain unchanged and maintain their predefined values during the training process?

2. In addition to the linear relationship shown in Fig 2, it would be helpful to see several other models to demonstrate the generalizability of this pattern.

3. The paper claims that the first-kind Chebyshev polynomial coefficients work best for recomposition, but limited rationale is provided for this choice.  Some theoretical analysis on why Chebyshev polynomials are particularly suitable would strengthen the motivation.

**Questions:**

Please see weakness section.

**Limitations:**

Please see weakness section.

---

> ### Author Rebuttal · Authors · 2024-08-07
>
> To 1: Details of recomposition
>
> Thank you for your feedback. In the revision, we will provide a more detailed explanation of the recomposition process and update Fig.3 in the paper to illustrate the initialization of different model architectures and the various training methods.
> In Fig.D of the PDF, we provide two examples. At the top of Fig.D, a 2-layer ViT is initialized with 3 components, and at the bottom, a 3-layer ViT is initialized with 4 components. Regardless of the training method, different model structures are initialized using Eq.9. As shown in Fig.D, for the "without constraint" training, Eq.9 is used solely for initialization. During training, the parameters of each layer ($W_l$) are updated independently, similar to a typical ViT, without adhering to Eq.9. In contrast, the "with constraint" training method updates the parameters of the component $A_n$ during training and maintains the polynomial coefficients unchanged, meaning the components remain shared across all layers, and each layer's parameters continue to satisfy Eq.9.
>
>
> To 2: More linear relationship examples for other models
>
> Thank you for the suggestion. The supplementary material provides results for MoCo and DINO models using the same process as described in the paper. Additionally, in Fig.B, we show the relationship between each layer's parameters and the corresponding layer index for MAE-B[1], BEITv2-B[2], and RoBERTa-B[3] models. It is evident that these models also exhibit a linear relationship between the parameters and the layer index. In the revision, we will include these results to further demonstrate the generalizability of this pattern.
>
> To 3: Chebyshev polynomials
>
> Thank you for your observation. In Tab.1 of paper, our experimental results demonstrate that Chebyshev polynomials outperform other polynomials in recomposition. This may be because Chebyshev polynomials are known to provide the best uniform approximation of continuous functions [4]. And each layer of the ViT model can be viewed as a function, the parameter matrix of each layer can be decomposed into components and represented as a weighted sum using Chebyshev polynomials. In the revision, we will expand on this theoretical motivation and explicitly highlight the desirable mathematical properties of Chebyshev polynomials.
>
>
> [1]Masked Autoencoders Are Scalable Vision Learners.
>
> [2]BEIT: BERT Pre-Training of Image Transformers
>
> [3]RoBERTa: A Robustly Optimized BERT Pretraining Approach
>
> [4]Trefethen, L. N. (2013). Approximation Theory and Approximation Practice. SIAM.

---

> > ### Comment · Reviewer_DK4A · 2024-08-09
> >
> > Thank you for your well-crafted rebuttal, which has effectively addressed the majority of the concerns I raised in my initial review. After carefully considering the comments from other reviewers and your responses, I stand by my original recommendation to accept this manuscript. The strengths of the paper remain clear, and I believe it makes a valuable contribution to the field. Congratulations on this solid piece of work.

---

> > > ### Author Response · Authors · 2024-08-09
> > >
> > > Thank you very much for your kind words and for taking the time to review our manuscript. We greatly appreciate your thoughtful feedback and are pleased that our responses have addressed your concerns. Your recommendation and positive comments are highly encouraging, and we are delighted that you find our work to be a valuable contribution to the field.

---

### Official Review · Reviewer_WjQY · 2024-07-13

**Soundness:** 3
**Presentation:** 3
**Contribution:** 3
**Rating:** 5
**Confidence:** 4

**Summary:**

Inspired by polynomial decomposition in calculus, the authors propose linearly decomposing the ViT model into a set of components during element-wise training. These components can be recomposed into differently scaled, pre-initialized models to meet different computational resource constraints. This decomposition-recomposition strategy offers an economical and flexible approach for generating diverse scales of ViT models for various deployment scenarios. Unlike model compression or training from scratch, which require repeated training on large datasets for different-scale models, this strategy reduces computational costs by training on large datasets only once. Extensive experiments demonstrate the method's effectiveness, showing that decomposed ViTs can be recomposed to achieve comparable or superior performance to traditional model compression and pre-training methods.

**Strengths:**

The authors successfully decompose and recompose the layers of transformers, achieving competitive performance across various component configurations, parameter settings, and datasets.

**Weaknesses:**

There are many variants of pure vision transformers, such as Swin and MobileViT, which significantly outperform traditional vision transformers. It would be valuable to retrain these models for scenarios requiring different model sizes and datasets.

**Questions:**

1.Can you demonstrate that your method is applicable to other well-known transformer variants such as Swin and MobileViT?

2.Can you compare the performance of your proposed models, decomposed and recomposed from DeiT-B, against other state-of-the-art models with the same parameter settings?

**Limitations:**

I shared my concerns in the weakness section.

---

> ### Author Rebuttal · Authors · 2024-08-07
>
> To 1: Swin Transformers and MobileViT
>
> Thank you for your question, which inspired us to explore the applicability of our method to Swin Transformers and MobileViT.
> Swin Transformers consist of four stages, where different stages have transformer layers with different resolutions, resulting in different components decomposed from layers across different stages. Due to time constraints, we focused on decomposing the layers in stage 3 because it has a larger number of layers. We visualized the relationship between the layer parameters and the corresponding layer index in stage 3. In Fig.A(a) and Fig.A(b) of PDF, we present the results for Swin-T and Swin-B, respectively. We observed a linear relationship between the parameters and the layer index, similar to that shown in Fig.2 in paper for plain transformers. Therefore, we adapted our method for Swin transformers by treating stage 3 as a plain transformer during decomposition. We have successfully decomposed one component from stage 3 of the Swin-Tiny model, achieving a result of 77.61\%. This is close to the Swin-Tiny's performance on ImageNet-1k, which is 81.2\%, and the initial results are promising. We will include the full experimental design and results in the revision.
>
> For MobileViT, in Fig.A(c) we also visualize the transformer layers across different ViT blocks. We observed that within each block, there is a linear relationship between the layer parameters and the layer index. However, since MobileViT has relatively fewer parameters, it may not be economical and efficient to decompose it. Given the linear relationship, it is still possible that the transformer layers in MobileViT could be initialized with pre-trained components. We will perform further experiments to validate this possibility and incorporate these results in revision.
>
>
> To 2: More comparative experiments
>
> Thank you for your question. For decomposition, as in Fig.4(a), our decomposed models outperform the original DeiT-B model with the same number of parameters under the same training conditions. Fig.4(b) further shows that our decomposition approach leads to better performing models in different parameter regimes compared to the SN-Net family.
>
> For the recomposition, we compared our recomposed models to different-sized DeiT models and other compression methods such as Mini-DeiT-B and Efficient-ViT. The parameters for these models were downloaded from their official sources and fine-tuned under the same training epochs as ours over 9 downstream tasks. The results in Fig.6 demonstrate the effectiveness of our method. In addition, we provide detailed numerical performance in the supplementary material (Tab.2).
>
> Furthermore, we conducted additional comparison experiments with the 12-layer ViT, based on the DeiT-Base structure, initialized with 4 components and trained using the "with constraint" method (28.7M parameters), compared to the pre-trained Swin-Tiny (28M parameters), as shown in Tab.A of the PDF. The results indicate that the model initialized with components performs comparably to the pre-trained Swin-Tiny on downstream tasks with a similar number of parameters. This further demonstrates the feasibility of using components to initialize ViTs of different scales. We will provide the full experimental results in the revision.

---

> > ### Comment · Reviewer_WjQY · 2024-08-13
> >
> > The authors address parts of my questions. With the current experiments, it is hard to agree that the proposed method outperforms recent transformer models in various sizes such as Inception transformer, multiscale vision transformer, cswin. So I will keep the current rating.

---

> > > ### Author Response · Authors · 2024-08-13
> > >
> > > Thank you very much for your thoughtful feedback. We greatly appreciate the time and effort you have put into reviewing our work. Your insights are invaluable, and we will take your suggestions into account as we plan to conduct further experiments to provide a more comprehensive evaluation.

---

### Author Rebuttal · Authors · 2024-08-07

We greatly appreciate the insightful and constructive feedback provided by the reviewers. We have responded in detail to the concerns and questions raised, and have attached a PDF with figures and tables. Below is a brief overview of the PDF:

1. Fig. A: Shows the parameters of each layer of the pre-trained Swin Transformer/MobileViT and their corresponding layer position relationships (to Reviewer WjQY).

2. Fig. B: Presents additional examples of linear relationships for other models, illustrating the parameters of each layer and their corresponding layer position relationship (to Reviewer DK4A).

3. Fig. C: Demonstrates the decomposition process using DeiT-Base as an example (to Reviewer caYK).

4. Fig. D: Shows the recomposition process and two training methods with two examples: the first is a 2-layer ViT initialized with 3 components, and the second is a 3-layer ViT initialized with 4 components (to Reviewer DK4A, caYK).

5. TABLE A: Contains comparative experimental results comparing the 12-layer ViT initialized with 4 components to the pre-trained Swin-Tiny (to Reviewer WjQY).

---

### Decision · Program_Chairs · 2024-09-25

**Decision:**

Accept (poster)

**Comment:**

The authors propose an parameter-efficient learning technique. The final score is 5767.

Overall, the author's rebuttal has won unanimous approval from the reviewers, and this article has made its own contribution. After reviewing the reviewers' comments, the author's responses, and the AC's discussion, we finally decided to accept this article, but based on the following two points:
1. The author needs to fulfill his promise and discuss the two articles raised by the reviewer caYK in the discussion section.

2. The author needs to change the tone of the conclusion and claim in the paper, because the current technology may not be able to surpass the following models in all sizes: Inception transformer, multiscale vision transformer, cswin